# On the effectiveness of GAN generated cardiac MRIs for segmentation

**Youssef Skandarani**[1,2]                    youssef_skandarani@etu.u-bourgogne.fr
**Nathan Painchaud**[3]                         nathan.painchaud@usherbrooke.ca
**Pierre-Marc Jodoin**[3]                       pierre-marc.jodoin@usherbrooke.ca
**Alain Lalande**[1,4]                          alain.lalande@u-bourgogne.fr

[1] *Laboratoire ImVIA, Université de Bourgogne Franche-Comté, Dijon France*
[2] *Cardiac Simulation & Imaging Software, Dijon France*
[3] *Department of Computer Science, University of Sherbrooke, Canada*
[4] *University Hospital of Dijon, France*

## Abstract

In this work, we propose a Variational Autoencoder (VAE) - Generative Adversarial Networks (GAN) model that can produce highly realistic MRI together with its pixel accurate groundtruth for the application of cine-MR image cardiac segmentation. On one side of our model is a Variational Autoencoder (VAE) trained to learn the latent representations of cardiac shapes. On the other side is a GAN that uses "SPatially-Adaptive (DE)Normalization" (SPADE) modules to generate realistic MR images tailored to a given anatomical map. At test time, the sampling of the VAE latent space allows to generate an arbitrary large number of cardiac shapes, which are fed to the GAN that subsequently generates MR images whose cardiac structure fits that of the cardiac shapes. In other words, our system can generate a large volume of realistic *yet labeled* cardiac MR images. We show that segmentation with CNNs trained with our synthetic annotated images gets competitive results compared to traditional techniques. We also show that combining data augmentation with our GAN-generated images lead to an improvement in the Dice score of up to 12 percent while allowing for better generalization capabilities on other datasets.

**Keywords:** GAN, CNN, Deep Learning, cine-MRI, Heart

## 1. Introduction

Acquiring large medical datasets is an expensive and time consuming endeavour, especially when they have to be manually annotated by experts. In this paper, we propose a combined Variational Autoencoder - Generative Adversarial Network (VAE - GAN) method for producing highly realistic cine-MR images together with their pixel-accurate groundtruth.

GANs (Goodfellow et al., 2014) are well-known for their ability to generate images whose distribution fits that of a predefined set of data. A subset of GANs are the image-to-image translation networks (Isola et al., 2017) that transform an input image from one domain, e.g. segmentation maps, to another domain, e.g. realistic images. Unfortunately, in the context of medical image segmentation, the ground-truth labels are either the result of a segmentation network (whose variety is limited to that of its input images) (Shin et al., 2018) or hand drawn (Abhishek and Hamarneh, 2019). In this paper, we overcome this problem by using a VAE that learns the underlying cardiac latent distribution and thus can generate an arbitrary large number of cardiac maps.

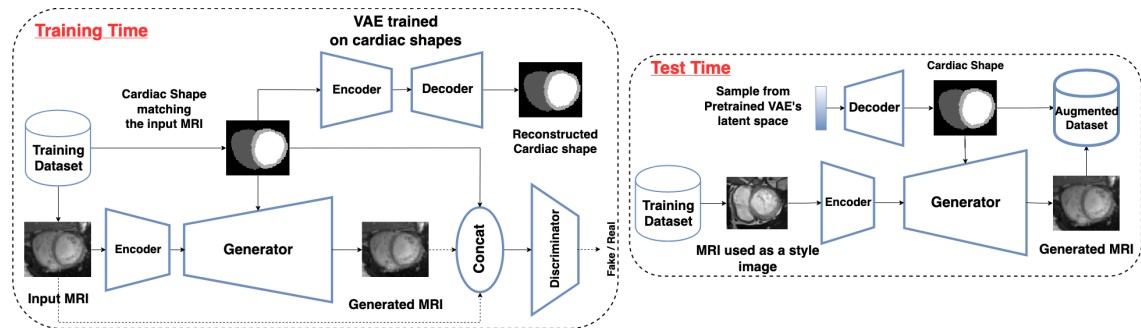

Figure 1: The proposed VAE-GAN MRI generation architecture.

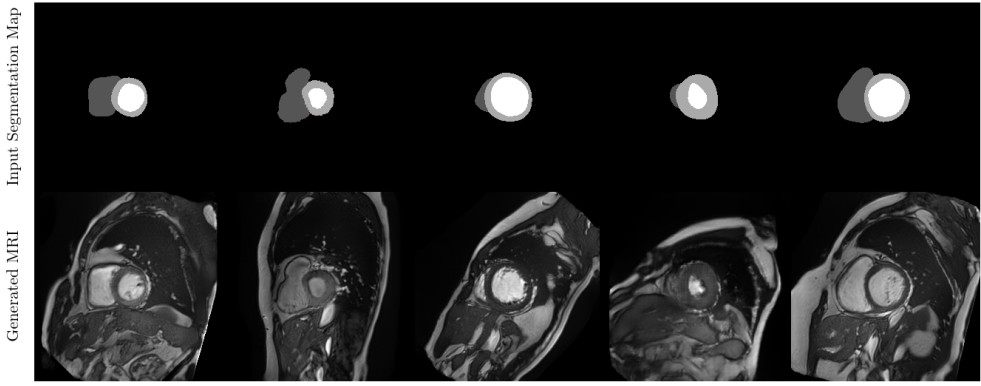

Figure 2: [Top] cardiac shapes generated by our VAE and [Bottom] synthetic GAN-generated cine-MR images conditioned on the cardiac shape.

## 2. Method

Our image generation pipeline uses a module called "SPatially-Adaptive (DE)Normalization" (SPADE)(Park et al., 2019) that is a conditional normalization layer within the generator (fig.1). The segmentation map is used as the condition for the SPADE module, which forces the generator to output an image whose structure fits that of the cardiac shape. In addition to the GAN is an anatomical variational autoencoder (VAE) (Painchaud et al., 2019) whose latent space can be sampled to produce anatomically valid cardiac shapes.

During training, the GAN is fed a MRI and, conditioned on its associated anatomical map, outputs a generated MRI, which are then both given to the discriminator. Simultaneously, the VAE is trained to learn the cardiac shapes' latent distribution. At test time, the system is fed a MRI and, conditioned on an arbitrary anatomical map generated by the VAE decoder, generates a MRI whose cardiac shape fits the anatomical map (fig.2).

## 3. Results and discussion

We trained and tested our framework on two cine-MRI datasets, namely ACDC (Bernard et al., 2018) (1902 training slices and 1078 testing slices) and the Sunnybrook Cardiac Data (Radau et al., 2009) (478 training slices and 236 testing slices), although the latter was re-annotated by an expert to match the segmentation specification of the ACDC dataset. These datasets contain cine-MRI images at end diastole and end systole and their associated

Table 1: Test Dice scores on ACDC and Sunnybrook datasets: [W/O DA] Training without data augmentation, [DA] Training with data augmentation, [Gen. ACDC] training on the ACDC-like 100k synthetic dataset, [Gen. Sunny] training on the Sunnybrook-like 100k synthetic dataset.

| Testing | Training | W/O DA | DA | Fine Tuning | |
| | | | | ACDC | Sunnybrook |
|---------|----------|--------|-----|------|------------|
| ACDC | ACDC | 0.844 | 0.854 | — | 0.406 |
| | Gen. ACDC | **0.849** | **0.888** | **0.908** | **0.784** |
| | Sunnybrook | 0.122 | 0.120 | 0.812 | — |
| | Gen. Sunny. | 0.336 | 0.588 | 0.832 | 0.488 |
| Sunnybrook | ACDC | 0.738 | 0.773 | — | 0.795 |
| | Gen. ACDC | **0.803** | **0.853** | 0.845 | 0.848 |
| | Sunnybrook | 0.776 | 0.798 | 0.820 | — |
| | Gen. Sunny | 0.773 | 0.816 | **0.845** | **0.874** |

expert segmentation for the left ventricular cavity, the myocardium and the right ventricular cavity. We trained our VAE-GAN separately on the ACDC and Sunnybrook datasets and then generated 100k synthetic images by sampling the anatomical VAE's latent space. We then trained an ENet CNN (Paszke et al., 2016) on the original datasets as well as on the 100k synthetic datasets. Table 1 summarizes the test results of ENet with and without fine tuning (training for a few epochs on the original datasets). Also, since our VAE-GAN can be seen as a sophisticated data augmentation, we trained the ENet with and without traditional data augmentation, i.e. random rotations, flips and shifts.

The ENet trained on the VAE-GAN generated datasets with data augmentation has Dice scores higher by 2 to 4 percent compared to an ENet trained on the original datasets (table 1). This gap is even more prominent when ENet is fine-tuned as it's Dice score increases by 6 to 10 percent. For instance, the ENet trained on the 100k synthetic Sunnybrook dataset with fine tuning on the original Sunnybrook dataset has a Dice score of 0.874 compared to only 0.776 when trained on the original Sunnybrook. This is a significant improvement considering that the used Sunnybrook training set contains only 478 2D slices.

Results also underline that our VAE-GAN coupled with data augmentation provides even better results (in table 1, 0.888 vs 0.849 and 0.853 vs 0.803). Moreover, the ENet trained on our VAE-GAN generated dataset has better generalization capabilities; the model trained on the synthetic ACDC dataset has a Dice score of 0.853 on the Sunnybrook dataset versus 0.773 when trained on the orginial ACDC dataset.

## 4. Conclusion

We presented a novel VAE-GAN cine-MRI cardiac generation model. This method has the sole ability of generating both a realistic cardiac MR images as well as its associated groundtruth. Results have been positive after training and testing on two datasets, especially when using data augmentation and fine tuning. Further investigations could be

done on the sampling from the VAE's latent space to help overcome the problem of class imbalance which may be present in certain medical imaging datasets. Time consistency of cine-MRI could also be explored to augment the generation process in future works.

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
