# OpenReview forum: "On the effectiveness of GAN generated cardiac MRIs for segmentation"
_MIDL.io/2020/Conference — MIDL 2020_

### Official Review · AnonReviewer3 · 2020-03-05
**GAN based synthesised method for cardiac data segmentation**

**Rating:** 2
**Confidence:** 4

**Review:**

The authors designed a GAN based synthesised method for cardiac data segmentation. I have some concerns regarding the results of this work: for instance, the authors proposed a complicated model but the segmentation results are quite similar or even slightly worse than the data augmentation. Also GAN based synthesise was proposed and widely tested before. The novelty of the study is quite limited.

---

### Official Review · AnonReviewer1 · 2020-03-12
**Interesting work - could be improved with some clarifications**

**Rating:** 3
**Confidence:** 4

**Review:**

The paper describes a method for GAN generation of cardiac MR images along with generated anatomical segmentations for these new images.  The authors then train a segmentation network and demonstrate improved results (in general) when the training set is enlarged with GAN-generated images and their (generated) segmentations.

The method described is interesting and should prove useful in the absence of large annotated datasets.  The results seem reasonably convincing although I have some questions.  Given the limitations of the short paper I think that this is an interesting  and reasonably well described work.  Specific comments to improve the paper are below:

 - The specific anatomical structures segmented are not named
 - The sizes of the datasets used should be mentioned, since these are clearly very important for the reader's understanding (and to save them searching the literature)
 -  The properties of the test sets are not described - from which set do they come and how large are they?   Validation data is also not described.
 - The term fine-tuning is used without explanation until late in the paper - the meaning of this term in the context of this work should be explained before presenting the results.
- (outside the scope of this short paper) It would be interesting to experiment with combining the two public datasets .  It would also be useful to see a graph of how the results improve as additional training images are added - is there a saturation point beyond which additional training images are not useful?  Are the generated images more useful if there was a larger number of images (i.e. more variability) in the dataset that was used to train the generator?

---

### Official Review · AnonReviewer2 · 2020-03-14
**Interesting results, worth to have a deeper discussion on the method**

**Rating:** 2
**Confidence:** 5

**Review:**

The authors combine a variational auto-encoder and a GAN to generate synthetic cine MR images along with the associated labels. They test their method on two well-known datasets showing good results.

Clarity: Unfortunately there is not enough space to discuss details about the method and potential limitations (see comments at the end). There are some parts of the results that are not commented. What is fine tuning?
Quality: Average. The paper would gain quality if there could be more discussion on the results. I would suggest to remove the data augmentation and fine tuning experiments, as it is not the goal of the paper, to gain place for explaining the limitations of the paper.
Significance: The authors address a relevant problem.

Pros:
- Reported accuracy is good

Cons:
- You use cine MR images. This images have a temporal component. How do you guarantee that the results are consistent in time? I mean does the image in time T is consistent with that one at T+1? If you are not addressing this, at least, it should be mentioned.
- Using a given dataset to generate a synthetic dataset and then use images from the original dataset for the testing (table 1) is not very challenging as a test. Most likely, there is bias.
- The experiments on data augmentation and fine tuning (which is not really commented) are somehow out of place. The point of the paper is to show the value of the synthetic data generation.
- The method tries to solve the problem of limited annotated data. However, GANs are methods that require a lot of data to be able to generate good results. The experiments are not sufficient to proof that it would not be the case in this setup and the authors don't really comment on it.

---

### Official Review · AnonReviewer4 · 2020-03-14
**Nice application of VAE+GAN architecture to generating synthetic cardiac cine-MRI**

**Rating:** 3
**Confidence:** 4

**Review:**

In this paper, the authors present a deep learning model that integrates a variational auto encoder (VAE) and a generative adversarial network (GAN) to generate synthetic images conditioned by synthetic labels. The method is novel, however I do have some concerns regarding the method to ensure the consistency between the MRI used as a style image and the generated cardiac shape.

The evaluation explores different experimental setups for two different open datasets available in cardiac cine-MRI (ACDC and Sunnybrook cardiac data) which makes the paper interesting to read. It would be interesting to know additional details about the data augmentation procedure used on the experiments. The observed gain in performance in the segmentation task seem to be quite large and probably significant, however comparison with alternative methods for generating synthetic cardiac-MRI images is missing.

---

### Meta-Review · Area_Chair1 · 2020-04-08
**MetaReview of Paper118 by AreaChair1**

**Rating:** 2

**Metareview:**

This paper describes a way to use VAE to produce more MR images, which in turn contribute to segmentation. Review comments are generally at borderline. The novelty and evaluation of the paper might be limited though.

**Paper Type:**

both

---

### Decision · Program_Chairs · 2020-04-11

**Decision:**

Accept

**Comment:**

Taking all information into account, it was determined that the paper was accepted based on its merit.